# Within-Person Modulation of Neural Networks following Interoceptive Awareness Training through Mindful Awareness in Body-Oriented Therapy (MABT): A Pilot Study

**DOI:** 10.3390/brainsci13101396

**Published:** 2023-09-30

**Authors:** Cynthia J. Price, Gunes Sevinc, Norman A. S. Farb

**Affiliations:** 1School of Nursing, University of Washington, Seattle, WA, USA; 2Department of Psychiatry, Massachusetts General Hospital, Harvard Medical School, Boston, MA, USA; gunes.sevinc@ardeaoutcomes.com; 3Ardea Outcomes, Halifax, NS, Canada; 4Department of Psychology, University of Toronto Mississauga, Mississauga, ON, Canada; norman.farb@utoronto.ca

**Keywords:** interoception, fMRI, neuroplasticity, neural networks, mindfulness, randomized controlled trial

## Abstract

Interoception, the representation of the body’s internal state, is increasingly recognized for informing subjective wellbeing and promoting regulatory behavior. However, few empirical reports characterize interoceptive neural networks, and fewer demonstrate changes to these networks in response to an efficacious intervention. Using a two-group randomized controlled trial, this pilot study explored within-participant neural plasticity in interoceptive networks following Mindful Awareness in Body-oriented Therapy (MABT). Participants (N = 22) were assigned to either 8 weeks of MABT or to a no-treatment control and completed baseline and post-intervention assessments that included subjective interoceptive awareness (MAIA) and neuroimaging of an interoceptive awareness task. MABT was uniquely associated with insula deactivation, increased functional connectivity between the dorsal attention network and the somatomotor cortex, and connectivity changes correlated positively with changes in subjective interoception. Within the MABT group, changes in subjective interoception interacted with changes in a predefined anterior cingulate seed region to predict changes in right middle insula activity, a putative primary interoceptive representation region. While the small sample size requires the replication of findings, results suggest that interoceptive training enhances sensory–prefrontal connectivity, and that such changes are commensurate with enhanced interoceptive awareness.

## 1. Introduction

Interoception is the process of sensing, representing, and appraising the body’s internal state [1]. Interoception is increasingly valued as a process of integrating homeostatic cues [2,3,4] that are arguably the primary motivators of human behavior [5]. Over the course of development, interoceptive signals become linked to conditioned appraisals of wellbeing and associated regulatory responses; as such, interoception serves as a critical determinant of both mental and physical health [6]. In accordance with its central role in promoting homeostasis, interoception relies on a network of viscerosomatic cortical regions, such as the posterior insula and supplementary somatosensory cortex [7], and may engage attentional control regions that are distinct from the frontoparietal network commonly implicated in externally directed attention [8,9].

In recent years, interoceptive awareness has become a popular explanatory mechanism within the clinical sciences due to its role in informing a person’s sense of wellbeing [10]. While ideally interoceptive signals motivate adaptive regulatory behavior, chronic stress may compromise this process by altering the availability or tolerability of interoceptive signals, which in turn may compromise the accurate interpretation of sensations to predict adaptive responses [11,12]. Neurally, such dysfunction may be reflected by dysregulation within the insula, with the posterior insula serving as a putative primary interoceptive representation cortex [7]. While the anterior insula is acknowledged as an allostatic network hub for polymodal sensory convergence [9], the posterior and middle insula is thought to specifically support interoception and influence the self-appraisal of affective symptoms [6,13]. In clinical conditions, dysfunction in this insula-based network is commensurate with the type of mental health distress: anxiety, for example, has been linked to increased insula activation [14,15], whereas depression may be linked to hypo-activation in the insula and nearby somatosensory cortex [16,17]. Affective distress can thus be understood as an insula-mediated conditioned response to aversive states, involving neurological dysfunction and a lack of internal regulation [18]. Furthermore, interoceptive awareness appears to be indicated by prefrontal activity, and in particular in the dorsal anterior cingulate cortex (ACC), whose activity predicts awareness during attention to interoceptive signals [19].

To the extent that maladaptive interoceptive conditioning underlies clinically significant mental health conditions, the cultivation of interoceptive awareness may de-automate this process and support the cognitive reframing critical for adaptive emotion regulation [6,20,21]. Attending to varied interoceptive responses to events affords novel appraisals of these events, promoting flexibility in self-appraisal that may constructively interfere with depressive or anxious interpretive habits [22,23,24]. Accordingly, Mindfulness-Based Stress Reduction (MBSR)—a validated clinical intervention that aims to cultivate interoceptive awareness—has been linked to enhanced connectivity between interoceptive representation regions and the prefrontal cortex (PFC) [25]. While this study explored cross-sectional differences between trainees and waitlist controls, a more recent investigation of within-participant effects of MBSR reported enhanced hippocampal connectivity to both the PFC and somatosensory cortices during the retrieval of extinguished fear memories, providing an initial implication of within-person interoceptive neuroplasticity following mindfulness training [26].

Despite these promising findings, no study has demonstrated within-person effects of interoceptive training on interoceptive networks during interoceptive attention. This gap in the literature is particularly concerning given the centrality of interoceptive brain networks in recent cognitive neuroscience models of psychological disorders [27]. Recent consensus statements increasingly recognize interoception in mental health, while acknowledging the scarcity of neural evidence supporting such theory [6]. While there are a growing number of neuroscience theories of interoception underlying psychopathology [27,28,29,30] and cross-sectional studies demonstrating aberrant interoceptive brain activity between clinical and healthy samples [31,32,33,34,35], little evidence supports the modulation of interoceptive networks as a mechanistic marker of clinical intervention.

Establishing within-participant, treatment-related plasticity in interoceptive brain networks is a critical step in testing interoceptive theories of mental health. While interventions featuring interoceptive training are regularly linked to changes in subjective interoception [36,37,38,39,40], such reports have not linked changes in subjective awareness to changes in underlying interoceptive brain networks. To demonstrate that reports of subjective interoceptive change are not a product of treatment-specific demand characteristics, and to establish that interoceptive brain networks serve as mechanistic targets of clinical intervention, a direct demonstration of treatment-related interoceptive network plasticity is needed.

Mindful Awareness in Body-oriented Therapy (MABT) is specifically designed to cultivate fundamental skills of interoceptive awareness. The intervention protocol includes multiple components including touch (to guide and support internal attention), mindfulness, and psychoeducation. A key component of the MABT is learning to engage in focused sustained interoceptive attention to increase awareness of sensations, including emotions, and to enhance somatic evaluation and appraisal [41]. MABT research indicates that interoceptive avoidance or dysfunction is common among individuals with distressing mental and physical health conditions [39,40,42,43]; MABT is thus delivered individually to allow for individualization of the teaching processes needed to promote successful development of interoceptive awareness skills and the integration of these skills into daily life [44]. When MABT is included as an adjunctive therapy, it is linked to reduced depression, emotion regulation difficulties, dissociation, and improved interoceptive awareness among women in substance use disorder treatment [39,40]. Longitudinal between-group effects link MABT to reduced craving and substance use, improved respiratory sinus arrhythmia (a psychophysiological indicator of emotion regulation), and importantly to improvements in subjectively-reported interoceptive awareness [40]. Improved subjective interoception has been identified as a mechanism underlying positive changes in wellbeing in response to MABT [43,45].

Here we describe the first randomized controlled neuroimaging study designed specifically to uncover evidence of plasticity in interoceptive neural networks in the context of a clinically effective interoceptive training (MABT). An initial cross-sectional study with this sample showed diffuse cortical deactivation during an active interoception vs. exteroceptive condition, while higher scores on the MAIA predicted that the anterior cingulate cortex (ACC) and left-lateralized language regions were spared from this deactivation [46]. In this study focused on potential MABT treatment effects, we examined the role of interoceptive training specifically on insula deactivation. We also examined whether interoceptive network connectivity in a moderately distressed community sample is enhanced in response to interoceptive training through MABT. The study aims were to: (1) examine interoception-related activity in the insula, based on an a priori anatomical RO1, (2) examine whether MABT alters functional connectivity in wide-scale brain networks, and (3) examine whether MABT alters functional connectivity with a task-derived, interoceptive seed region in the ACC.

## 2. Materials and Methods

### 2.1. Design Overview, Ethics Statement, and Experiment Power

To examine the causal influence of interoceptive training on neural networks supporting interoception, a two-group randomized controlled trial was employed. Participants were randomly assigned to the MABT intervention or to a no-treatment control condition. Study participation involved the administration of a set of questionnaires and neuroimaging protocol at two timepoints: a baseline assessment and a post-intervention assessment (at 10-week follow-up). Participants were remunerated $75 for completion of each assessment. Those assigned to MABT were offered eight 90-min weekly MABT sessions that were individually delivered by one of two therapists trained in the MABT protocol. All study procedures were approved by the Institutional Review Board at the University of Washington, in accordance with the Helsinki Declaration of 1975. All participants provided written informed consent. This clinical trial was registered with ClinicalTrials.gov (NCT03583060) and pre-registered with the Open Science Framework (OSF; https://osf.io/y34ja, accessed on 4 December 2018).

Given the centrality of enhanced interoception in the mindfulness literature and the prior observation of large effects of MABT on subjective interoception [40], the decision was made to power this exploratory study to detect only medium to large Group × Time interaction effects (f > 0.3). Power analysis conducted in G*Power software (version 3.1.9.6) suggested that an N = 25 would only be sufficient to detect such effects with 80% power. This small sample size was sufficient for testing a focal hypothesis that anticipates a large effect of MABT training relative to the control, but is admittedly underpowered to detect weaker downstream effects on clinical symptoms or other mechanistic markers.

### 2.2. Participants

Twenty-five healthy individuals with self-reported elevated stress were recruited through advertisements in a local newspaper and through the University of Washington (UW) research volunteer website and flyers posted on the UW campus. The research coordinator screened interested potential participants for study eligibility. Inclusion criteria were: (1) adult over 18 years of age; (2) Perceived Stress Scale [47] scores indicating moderate stress levels; (3) naive to mindfulness-based approaches (no prior experience), (4) agrees to forgo (non-study) manual therapies (e.g., massage) and mind–body therapies (e.g., mindfulness meditation) for the study period (baseline to post-intervention); (5) fluent in English; (6) can attend MABT and research assessments; and (7) right-handed (for uniformity of neuroimaging results). Exclusion criteria were: (1) lifetime diagnosis of mental health disorder; (2) unable to complete study participation (including planned relocation, pending inpatient treatment, planned extensive surgical procedures, etc.); (3) cognitive impairment, assessed by the Mini-Mental Status Exam (MMSE) [48] if demonstrated difficulty in comprehending the consent; (4) use of medications in the past 30 days that affect hemodynamic response; (5) lifetime head injuries or loss of consciousness longer than 5 min; (6) currently pregnant; or (7) contraindications for MRI, e.g., claustrophobia, metal objects in body, etc.

Fifty-seven people responded to recruitment efforts and were screened for eligibility. Of these, 23 were eligible and interested in study participation. The final sample was made up of 22 individuals, due to the loss of one participant who withdrew from the study. The final sample included 11 participants in each experimental condition: MABT and a no-treatment control group. Participants’ ages ranged from 18 to 62 years (mean = 36.1); 20 participants self-identified as Caucasian, 1 as African American, and 2 as Hispanic; and 11 identified as male and 11 as female. Their highest education levels were high school (n = 5), 2 years of college (n = 2), bachelor’s degree (n = 8), and master’s degree or higher (n = 7). At baseline, participants’ screening scores indicated overall moderate severity for anxiety and depression symptoms: mild (n = 6), moderate (n = 13 for anxiety; n = 14 for depression), and severe (n = 3 for anxiety; n = 2 for depression). All participants assigned to receive the intervention completed at least 75% of the sessions (>6 sessions); specifically, eight completed 8 sessions, two completed 7 sessions, and one completed 6 sessions. Intervention completion (>75%) is critically important for exposure to all intervention components, because skills are taught sequentially over time and prior research demonstrates that receiving 6 or more MABT sessions brings about significantly better health outcomes than receiving fewer than 6 sessions [40].

### 2.3. Measures

Interoceptive Awareness. The Multidimensional Assessment of Interoceptive Awareness (MAIA) version 2 [6,49] was used to examine subjective interoception. The planned analyses used the MAIA total score following established scoring instructions. As the MAIA is composed of eight subfactors (Attention Regulation, Body Listening, Emotional Awareness, Noticing, Not Distracting, Not Worrying, Self-Regulation, and Trusting), exploratory analyses were also applied to the MAIA subfactors to establish whether training effects were driven by particular aspects of the multifaceted scale.

Symptom Burden. Participant symptom burden was assessed using the well-validated Patient Health Questionnaire: Somatic, Anxiety and Depression Symptom Scales (PHQ-SADS) [50], which was developed to efficiently evaluate the depression, anxiety, and somatization domains of affective symptom burden [50]. Specifically, the symptom burden measures included the following: the PHQ-9 (depression) [51], the GAD-7 (anxiety) [52], and the PHQ-15 (somatic symptoms) [53]. The scales were all administered with standard instructions and traditional computation of scale scores. Additionally, perceived stress was assessed as an additional indicator of participant symptom burden using the Perceived Stress Scale (PSS) [47].

To reduce the number of formal statistical tests, symptoms were summarized via an a priori planned factor analysis to compute a single factor score for the symptom burden measures derived from the PHQ-SADS (Depression, Anxiety, and Somatic Symptoms) and perceived stress (PSS) scales. The adequacy of a single factor to capture the variance in these scales was first assessed using the ‘paran’ library in the R statistical environment, which implements Horn’s test of Principal Components/Factors [54]. This analysis confirmed that 1 dimension was sufficient to capture variance across the 4 subscales at greater-than-chance levels. The 4 subscales were then reduced to a single set of factor scores using the factor analysis function ‘fa’ in the R ‘psych’ library [55]. The resulting factor score accounted for 64% of the scale variance; all 4 subscales’ factor loadings were greater than 0.60.

Mindfulness Skills. To examine self-appraisal skills that involve the ability to take a non-judgmental and observational perspective (i.e., viewing one’s experiences from a psychologically distant or third-person perspective), we used the Decentering subscale from the Experiences Questionnaire [56] and the Act without Judgement subscale from the Five Factor Mindfulness Questionnaire [57].

### 2.4. Intervention: Mindful Awareness in Body-Oriented Therapy

MABT is a manualized protocol developed by the first author (C.J.P.). The protocol is designed to teach fundamental skills of interoceptive awareness using a combination of manual, psychoeducation, and mindfulness approaches. The protocol uses an incremental approach to teach interoceptive awareness and related take-home skills in three distinct stages. Stage 1 (sessions 1–2) focuses on identifying body sensations; Stage 2 (sessions 3–4) focuses on learning strategies for interoceptive awareness; and Stage 3 (sessions 5–8) focuses on developing the capacity to sustain interoceptive awareness as a mindful process to facilitate self-acceptance, understanding, and appraisal of interoceptive experiences [41].

### 2.5. Data Collection Procedures

At the baseline and post-intervention assessments, the participants were first administered a set of self-report measures and then went through a functional imaging protocol that involved an interoceptive awareness meditation task.

### 2.6. The Interoceptive/Exteroceptive Attention Task

Undertaken in the scanner, this 5.5-min task compared behavioral tracking of the respiratory cycle (active interoception) to tracking of a visual stimulus (active exteroception), and is described in detail in a prior publication [46].

### 2.7. Interoceptive Awareness Meditation Task

In the scanner, participants listened to a 2.5 min audio-guided interoceptive awareness meditation before data acquisition, which directed them to place a hand on their chest and direct mindful attention to the inner space of the chest underneath their hand. Then, participants were instructed to sustain attention to inner body awareness with their eyes closed over a 3-min period of data acquisition. This procedure was repeated across two runs and yielded a total of 6 min of sustained attention data.

### 2.8. Functional Imaging Data Acquisition

Neuroimaging was performed using a 3T Philips Achieva scanner (Philips Inc., Amsterdam, The Netherlands) at the Diagnostic Imaging Sciences Center, University of Washington. Imaging began with the acquisition of a T1-weighted anatomical scan (MPRAGE) to guide normalization of functional images (~6 min) with TR = 7.60 ms, TE = 3.52 ms, TI = 1100 ms, acquisition matrix = 256 × 256, flip angle = 7°, shot interval = 2530 ms, and 1mm isotropic voxel size. Functional data were acquired using a T2∗-weighted echo-planar-imaging (EPI) sequence with TR = 2000, TE = 25 ms, flip angle α = 79°, field of view = 240 × 240 × 129 mm, 33 slices, and a voxel size of 3 × 3 × 3.3 mm with 3.3 mm gap. Button presses were registered using a 2-button MR-compatible response pad.

### 2.9. Image Processing

Preprocessing was performed using the fMRI Prep pipeline 20.0.6 [58] (see Appendix A for full details of preprocessing steps). Briefly, preprocessing consisted of realignment and unwarping of functional images, slice timing correction, and motion correction. The functional images were resliced using a voxel size of 2 × 2 × 2 mm and smoothed using a 6 mm FWHM isotropic Gaussian kernel. ARtifact detection Tools (ART) were used to detect frames with fluctuations in global signal and motion outliers. Intermediate level thresholds were used to reject 3% of the normative sample data. The frames with motion outliers that exceeded 0.9 mm or fluctuations in global signal > 5 standard deviations were considered outliers.

A component-based noise correction method (CompCor) [59] was used to address the confounding effects of participant movement and physiological noise. Structural images were segmented into cerebrospinal fluid (CSF), white matter, and grey matter. The principal components related to the segmented CSF and white matter were extracted and included as confound regressors in a first-level analysis along with movement parameters and breathing rate. The data were linearly detrended and band-pass filtered to 0.008–0.09 Hz without regressing the global signal.

### 2.10. Preliminary Analyses

Preliminary analyses were performed prior to addressing the research hypotheses. First, we analyzed the self-report data to characterize subjective training effects and validate the perceived impact of the MABT intervention. The self-reported outcome measures (interoception, symptoms, and mindfulness skills) were entered separately into between-groups fixed effects linear models to test for baseline differences. To assess treatment-specific changes, outcome scores were then modeled using linear mixed-effects models with restricted likelihood estimation using the ‘lme4’ library in the R statistical programming environment [60]. Group (MABT vs. control) × time (baseline vs. post-intervention) interactions were modelled as fixed effects with participant ID as a random error term to account for the within-subjects design.

Second, we established a priori regions of interest (ROIs). Two ROIs were based upon prior research. First, we used anatomical masks for the bilateral insula and the putative primary interoception cortex, which were created in Farb et al. (2013) [8]; these masks allowed for parcellation of the insula into specific gyri for a more nuanced analysis of activation differences between anatomical subregions. Second, we identified an anterior cingulate cortex (ACC) ROI identified as a MAIA covariate during interoception in prior research [46]. The insula and ACC are both theoretically consistent ROIs as they serve as the afferent and efferent hubs of the brain’s salience network [61] and are implicated as neural correlates of interoceptive awareness [62,63].

### 2.11. Neuroimaging Analysis

To assess training-related changes to activity within the insula (Aim 1), activation maps from a previously reported [46] whole brain analysis were used, which contrasted interoceptive and exteroceptive attention (a breath tracking task vs. a visual tracking task). The activation signal from both task conditions was extracted separately for each of the eight insula ROIs in each of the two hemispheres (16 ROIs in total; Figure 1a). The extracted activation was entered into a mixed-model multilevel model analysis using the ‘lme4’ library in the R statistical programming environment [60]. Group (MABT vs. control) × time (baseline vs. post-intervention) × condition (interoception vs. exteroception) × hemisphere (left vs. right) * region (the eight insula ROIs) were modelled as fixed effects with participant ID as a random error term to account for the within-subjects design. All analysis code for this and subsequent aims is available on the Open Science Framework (https://osf.io/ctqrh/ (accessed on 4 December 2018)).

Aims 2 and 3 focused on data from the Interoceptive Awareness Meditation Task, a sustained attention procedure that did not require behavioral responses. To assess changes in functional connectivity (Aim 2), we used a generalized Psycho-Physiological Interaction (gPPI) approach implemented in the CONN toolbox v.18b [64]. Combining data from both assessment timepoints, the analysis regressed the timeseries activity for each voxel on three parameters: (i) the mean timeseries within the ACC seed region, (ii) a vector contrasting timepoint levels (baseline vs. post-intervention), and (iii) the critical PPI regressor, which is the interaction between the seed timeseries and the time contrast vector. The resulting PPI map revealed the extent to which connectivity with the seed region varied as a function of time, with a single timepoint × connectivity interaction (PPI) map generated for each participant. PPI maps were entered into a second-level analysis with group (MABT vs. control) as a between-subject factor. The group level contrasts used a familywise error correction for a cluster significance of *p* < 0.05, given an uncorrected voxel height threshold of *p* < 0.001. The CONN toolbox (v.18b) provided Dorsal Attention (4 ROIs), Salience (7 ROIs), and Default (4 ROIs) network seeds, which were originally defined by CONN’s ICA analyses of the 497-subject HCP dataset. For each network, the average activity across voxels from all ROIs within the network was used as the seed timeseries regressor.

To assess interoceptive seed region connectivity (Aim 3), the task-derived ROI within the ACC (1205 voxels at 2 × 2 × 2 mm resolution; peak voxel at MNI coordinates x = −4, y = 12, z = 36) was used to further explore connectivity changes [46]. A gPPI model was used to assess changes in ACC functional connectivity strength across the two assessment timepoints (baseline vs. post-intervention), following the same methodology as above.

To address the overall study aims, group (MABT vs. control) × time (baseline vs. post-intervention) interaction analyses were employed to explore the effect of interoceptive training on: (i) self-reported interoceptive sensibility; (ii) interoception-related activity in the insula, an a priori anatomical ROI; (iii) well-characterized intrinsic connectivity networks; and (iv) functional connectivity with a task-derived ROI. We performed follow-up analyses to explore the relationship between neural changes and participant subjective change on the MAIA using change scores (post-intervention—baseline). Reported parameter estimates are standardized betas and 95% confidence intervals to facilitate comparison of effects. Simple effects of time within the MABT group only were also explored to maximize sensitivity to training effects.

The visualization of brain networks and seed region was performed using MRIcron software (version 1.0.20190902) [65]. All connectivity brain figures were generated using a combination of the CONN toolbox v.18b [64] and the wb_view visualization platform from the Connectome Workbench (WB) brain visualization and analysis software for brain images [66]. Data plots were generated using the ‘ggplot2’ library in R [67]. Complete R scripts and the dataset for tables, graphs, and follow-up analyses are available on the Open Science Framework (https://osf.io/ctqrh/ (accessed on 1 September 2023)).

## 3. Results

### 3.1. MABT Reduces Interoception-Related Insula Deactivation

Prior research revealed task-related deactivation across many cortical regions in a well-controlled comparison of exteroceptive (visual) attention vs. interoceptive (respiratory) attention (Farb et al., 2023 [46]). An insula ROI analysis of the group × time interactions with experimental condition (exteroception vs. interoception), laterality (left vs. right insula), and gyrus (eight anatomical subregions; Figure 1a) revealed group × time interactions with condition (β = 0.41, 95% CI [−0.08, 0.91], *p* < 1 × 10^−4^) with no evidence of moderation by laterality or specific gyrus (Figure 1b). Follow-up analyses suggested that the insula interaction was driven by a decreased distinction between exteroception and interoception in the MABT group (β_baseline_ = 0.48, β_post-intervention_ = 0.37) but an increased distinction in the control group (β_baseline_ = 0.21, β_post-intervention_ = 0.65).

### 3.2. MABT Increases Interoceptive Sensibility

No baseline differences were observed between study groups on any of the self-report measures. A group x time interaction was observed for self-reported interoceptive sensibility on the MAIA scale (β = 1.10, 95% CI [0.39, 1.82], *p* = 0.003) with fixed effects explaining 33.6% of all variance in MAIA scores (Figure 2a below). Simple effects analyses suggested that this interaction was driven by a significant increase in sensibility in the MABT group (β = 1.34, 95% CI [0.78, 1.90], *p* > 0.001) with time-related changes explaining 46% of MAIA variance, but little evidence of change in the control group (β = 0.41, 95% CI [−0.08, 0.91], *p* = 0.096) with time-related changes explaining only 4.3% of the variance. Exploratory analyses of MAIA subscales suggested that the same group × time interaction pattern was observable for all subscales, although interactions for the Emotional Awareness, Not Worrying, and Trusting subscales were not significant at an uncorrected *p* < 0.05 (Appendix A).

None of the symptom burden or mindfulness skills measures showed evidence of training-specific effects, although Decentering increased and Anxiety and Stress both decreased significantly as a main effect of time across the intervention period. As only MAIA scores were significantly impacted by training, only the MAIA total score was used as a covariate in subsequent analyses.

### 3.3. MABT Increases Insula Connectivity with the Dorsal Attention Network

A whole brain analysis of the group × time interaction using the dorsal attention network seeds (Figure 2b) yielded a significant cluster at in the right middle insula (Figure 2c), with MNI coordinates of x = 42, y = 08, z = 10 (k = 189, pFWE = 0.0149, beta = 0.05, t(21) = 5.26). Compared to the controls, MABT participants showed enhanced effective connectivity between the dorsal attention network and a cluster insula during the interoceptive awareness meditation task. Follow-up analysis revealed a positive correlation between increases in the insula–dorsal attention network connectivity and increases in MAIA scores (r = 0.589, 95% CI [0.22, 0.81], df = 20, *p* = 0.004) (Figure 2d).

### 3.4. MABT Increases ACC Connectivity with the Somatosmotor Cortex

A whole brain gPPI analysis of the changes in functional connectivity (baseline vs. post-intervention) using MAIA change scores as the regressor and the ACC as the seed (Figure 3a) failed to identify significant connectivity changes across groups. However, within the MABT group, the analysis yielded a significant cluster in the left somatomotor cortex of the pre- and post-central gyri, with MNI coordinates of x = −56, y = −16, z = 34 (k = 187, pFWE = 0.0013, Figure 3b).

Follow-up analyses across all participants revealed a positive correlation between increases in the ACC–somatomotor cortex connectivity and increases in MAIA scores (r = 0.583, 95% CI [0.21, 0.81], df = 20, *p* < 0.005) (Figure 3c). Unlike the pattern of insula connectivity with the dorsal attention network (DAN) described above, this effect was consistently found within the MABT group from which the ROI was identified, with a near-perfect correlation observed in MABT participants’ MAIA scores and ACC–somatomotor cortex connectivity (r = 0.973, 95% CI [0.90, 0.99]); there was no evidence of such a relationship within the control group (r = 0.067, 95% CI [−0.55, 0.64]).

Additional exploratory analyses suggested no additional associations between the DAN–insula and ACC–somatosensory connectivity scores and self-reported affective symptoms, decentering, non-judgment, or perceived stress (Appendix A). The DAN–insula connectivity scores were significantly associated with the MAIA subscales of Attention Regulation, Emotional Awareness, Self-Regulation, Body Listening, and Trusting. The ACC–somatosensory connectivity scores were significantly correlated with the subscales of Noticing, Attention Regulation, Self-Regulation, and Body Listening. However, the two regions did not show significant differences between their MAIA subscore connectivity profiles (Appendix A).

## 4. Discussion

This study provides an initial within-participant demonstration that interoceptive training improves neural markers of interoceptive function. Participants randomly allocated to MABT demonstrated within-person changes to brain activity and connectivity within intrinsic brain networks. We observed (i) training-related reductions in insula deactivation during a breath monitoring task, (ii) increased connectivity between the dorsal attention network (DAN) and right middle insula with study-wide changes in subjective interoception (MAIA scores), and (iii) increased connectivity between the anterior cingulate cortex (ACC) and left somatomotor cortex as a function of subjective interoceptive change within the MABT group.

Demonstration of training-related changes to interoceptive brain networks is of great importance, given that the clinical efficacy of validated mindfulness-based interventions (MBIs) is often attributed to their ability to reconfigure participants’ interoceptive attention and appraisal habits. While MBIs are regularly linked to changes in subjective interoception [36,37,38,39,40,68,69], only one prior study [70], involving an uncontrolled MBI group, has shown brain changes using fMRI that are linked to improved interoceptive awareness.

This sample of moderately stressed, community-dwelling adults receiving MABT demonstrated training-specific increases in subjectively reported interoceptive awareness (MAIA scores), replicating and extending effects observed in more acutely distressed clinical populations [39,40,71]. There were, however, no changes in other measures indicating either symptom burden or the more general mindfulness skills of decentering and non-judgment. The lack of such changes may be due to the relatively low symptom burden at baseline in this cohort, coupled with a lack of power to detect weaker effects in a small sample. Conversely, these findings speak to the large effect of MABT on participants’ subjective appraisals of interoceptive awareness, relative to the more general demand characteristics of being allocated to an active treatment group. The question we then posed was whether training-related appraisals of interoceptive change were corroborated by changes in neural networks.

To this end, we first investigated interoception-related activity in the insula, and connectivity patterns in well-characterized neural networks during sustained attention to interoceptive signals. Insula deactivation was moderated by MABT training, clarifying the findings from an earlier within-participant analysis showing that active interoception (relative to exteroception) drove cortical deactivation [46]. The prior study suggested that greater self-reported interoceptive awareness (MAIA scores) reduced deactivation within the ACC and left-lateralized language regions, but the cross-sectional nature of the MAIA regressor contraindicated making any causal inference. Here, we show that random assignment to interoceptive training via MABT was also sufficient to mitigate the deactivation effect within the insula, sparing this region from the widespread deactivation observed during the breath monitoring task. This finding provides initial causal evidence that this sparing of interoceptive regions may be an acquired attentional skill. Further, the extent of the mitigation effect was not significantly related to changes in self-reported awareness (MAIA scores), unlike the connectivity results below, wherein the ACC seed ROI was previously identified by its association with MAIA scores in the same deactivation context. Speculatively, the ACC, rather than the insula, may be a more reliable correlate of self-reported interoceptive change during the deployment of interoception, though much more work is needed to test this idea.

In a whole brain analysis of connectivity to the functional connectivity to DAN, MABT training was linked to increased connectivity with a single region: the middle right insula, a putative conduit of interoceptive integration from the primary interoceptive representation cortex in the posterior insula [7,72] to the PFC. This finding provides initial within-participant validation of prior cross-sectional studies, which linked MBIs to greater connectivity between the posterior insula and middle insula region [25], and greater insula–dorsal PFC connectivity [73]. Consistent with the interpretation that increased insula–PFC connectivity underlies greater subjective interoceptive experience, DAN–insula connectivity increases were positively related to changes in subjectively reported interoceptive awareness. Furthermore, the relationship between DAN–insula connectivity change and subjective interoception (MAIA) was not driven purely by between-group differences, as the relationship was also evident within the control group alone. Functional connectivity between the DAN and insula may therefore serve as a broader indicator of interoceptive awareness even outside of training contexts, although more research will be needed to verify this claim. Other neural networks did not show evidence of training-specific effects, suggesting that enhanced dorsal–PFC connectivity to sensory cortices was the primary neurophenotype of altered attentional control following interoceptive awareness training.

Another line of investigation examined training-related connectivity changes in an ACC seed region previously linked to interoceptive accuracy in heartbeat detection [62]. In the current sample, this region was again linked to interoceptive awareness in a localizer task that is the subject of a separate report [46]. Analysis within the MABT group revealed training-specific functional connectivity increases between the ACC and the somatomotor cortex. The somatosensory cortex is functionally coupled with the posterior insula to constitute a body awareness network [8,74,75], which is again commensurate with the hypothesis that mindfulness training may leverage improved attentional control in the PFC to increase the availability of interoceptive signals to conscious reflection and control. This finding is in line with previous reports of interoceptive awareness’ role in facilitating reappraisal [76], and corroborates the role of interoception in emotional experience. However, the relationship between ACC–somatomotor connectivity and MAIA scores was not evident within the control group, suggesting this relationship may be specifically linked to interoceptive training rather than being a general indicator of interoceptive awareness. It should be noted that in a prior report [46], lower MAIA scores were associated with greater deactivation of the ACC during breath-focused attention. This ‘flattening’ of the activation profile may have limited the ability to observe connectivity during sustained interoceptive attention as reported here, as illustrated by a lack of relationship between ACC–somatosensory cortex connectivity and changes in MAIA score in Figure 3c. While these follow-up analyses are intriguing, the small sample size precludes any definitive judgments as to the validity of these connectivity markers as biomarkers of subjective interoception.

Overall, the results suggest that MBIs may enhance coupling between PFC attentional control regions and sensory regions dedicated to somatosensory and viscerosomatic representation. Despite a small sample size, patterns of enhanced connectivity were associated with increased subjective reports of interoceptive awareness. Attending to sensory domains has previously been shown to increase PFC connectivity with sensory cortices (cerebral cortex), and multiple cross-sectional studies support the idea that MBIs are linked to enhanced PFC–sensory integration [77,78]. Such findings are essential for validating popular neural models of psychopathology, as training-related within-participant evidence of functional plasticity in interoceptive networks is largely absent from the research literature, despite insula activity being dubbed as ‘the hidden island of addiction’ [29] or as a biomarker of anxious cognition [79]. It is therefore encouraging to see that subjective response to focused interoceptive training produces at least initial evidence of modulation in these same networks. Further research is needed to establish whether within-participant changes in interoceptive representation can be shown to mediate treatment response in clinical samples, in keeping with interoceptive theories of mental health.

Also of note is the identification of the somatosensory cortex in addition to the insula as a target of interoceptive training. This finding is in keeping with a growing literature suggesting that interoception is served not purely by viscerosomatic processing in the insula, but also by sensory receptors in the skin, which is largely mapped in the somatosensory rather than insula cortex [80]. The role of somatosensory processing in interoception is supported by studies of patients with insular lesions, who were able to detect changes in heart rate unless the skin on the chest surface was also anesthetized [74]. The finding that interoceptive training targets body representation in a multifaceted way helps to broaden the definition of interoception to include all sensory representations that provide information about the body, rather than limiting definitions to visceral sense receptors alone. This finding is consistent with recent work on depression relapse vulnerability [17] which similarly related depression symptom burden and relapse risk across both the insula and somatomotor regions.

These overall findings are also noteworthy and consistent with the theorized mechanism of interoceptive training programs such as MABT. Rather than simply increasing the strength of interoceptive signals, MABT aims to build the capacity for increased awareness, acceptance, and emotional processing of all facets of internal experience (reflected on items measured by the MAIA). A primary focus of MABT is to develop the capacity to sustain mindful attention to the body, an attentional skill that can facilitate new and increased awareness or insight about oneself or a situation and can lead to adaptive self-appraisal processes [41,81]. Increased insight about oneself is a key psychological process that informs wellbeing in mindfulness practice [82]. Consistent with literature outlining the role of interoception for regulation [83], interoceptive awareness can be understood as critical for emotion processing that can be consciously influenced by thought-based regulation strategies, which are mediated by the medial and lateral prefrontal cortex [84,85].

There are several limitations of this study that need to be noted, but also considered with the context of this being an initial pilot study. As mentioned above, power was limited due to the small sample size, such that only large effects were likely detectable. In addition, this was a relatively healthy high-functioning sample and the related lack of high symptom burden served as a floor effect, limiting the ability to capture meaningful clinical change. Mindfulness skill measures of decentering and non-judgement showed baseline to post-intervention improvement across both groups; but the small sample size likely limited the ability to detect differences in change between the groups.. Notably, the decentering and non-judgment measures are more general mindfulness skill indicators; given the focus of MABT on interoceptive awareness skill development it is not surprising that these measures did not show the level of change seen on the MAIA. Conversely, it is possible that interoception effects on the MAIA are less generalizable to other MBIs due to the uniquely targeted focus on the development of interoceptive awareness skills in MABT. It should also be noted that the sparing of deactivation may be more typical of a treatment response for depressive hypoactivation than anxious hyperactivation within the insula [14,15,16,17]; this subclinical sample, with mixed symptoms of both depression and anxiety, may therefore not be representative of treatment response in more predominantly anxious samples. Despite these overall limitations, this study highlights positive within-person training effects on interoceptive brain networks in response to MABT, pointing to the critical importance of larger studies with more diverse populations, including both clinical and non-clinical samples, to better understand the role of interoceptive awareness in mindfulness training. Likewise, we suggest the use of additional objective measures to better understand the relationships between physiological and mental health indices. Within-participant group × time designs are not typical in neuroimaging investigations of mindfulness training; as such designs are much more powerful than cross-sectional approaches, we recommend this design approach for future neuroimaging mindfulness research.

## 5. Conclusions

This study shows initial evidence for within-participant training-related plasticity in interoceptive networks. Reduced insula deactivation and increases in connectivity between the DAN and the middle right insula, and between the ACC and the somatosensory cortex, highlight the potential for quantifying the effects of interoceptive awareness training on a biological level. Connectivity changes were consistently associated with increases in self-report interoceptive awareness on the MAIA, suggesting that insula–prefrontal connectivity mediates greater subjective access to interoceptive information. While further research is needed, the present findings support the theories of enhanced interoceptive processing as a mechanism of MABT.

## Figures and Tables

**Figure 1 brainsci-13-01396-f001:**
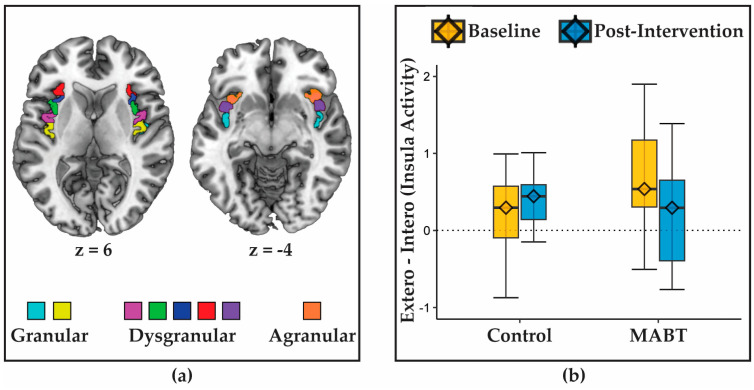
Effects of MABT training on interoception-related brain activity in a priori insula regions of interest (ROIs). (**a**) Eight insula ROIs as defined in Farb et al. (2013) [8] were selected as regions potentially sensitive to MABT training; (**b**) no ROI-specific effects of training were observed on the contrast of exteroception–interoception, but a general interaction effect was observed, such that the deactivation effect was reduced in the MABT group relative to the control group.

**Figure 2 brainsci-13-01396-f002:**
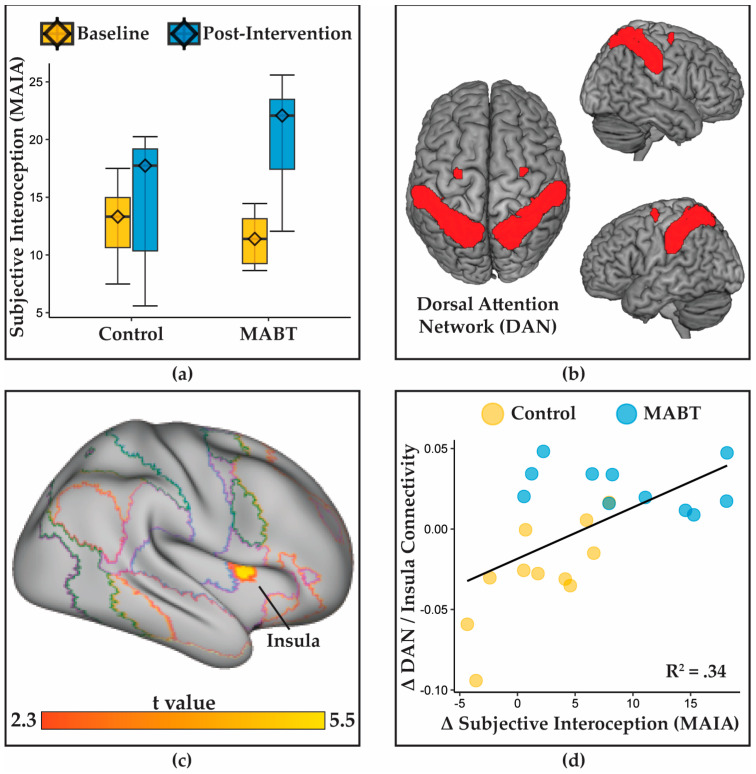
Effects of MABT training on subjective interoception and connectivity to the a priori anterior cingulate cortex (ACC) seed ROI: (**a**) group-specific training effects of MABT on subjective interoception (MAIA scores); (**b**) the dorsal attention network mask provided in [64]; (**c**) the right insula region demonstrating increased connectivity to the ACC ROI as a function of group (MABT > control); (**d**) the relationship between changes in subjective interoception and changes in ACC/insula connectivity. The line of best fit and R^2^ value are calculated across all participants.

**Figure 3 brainsci-13-01396-f003:**
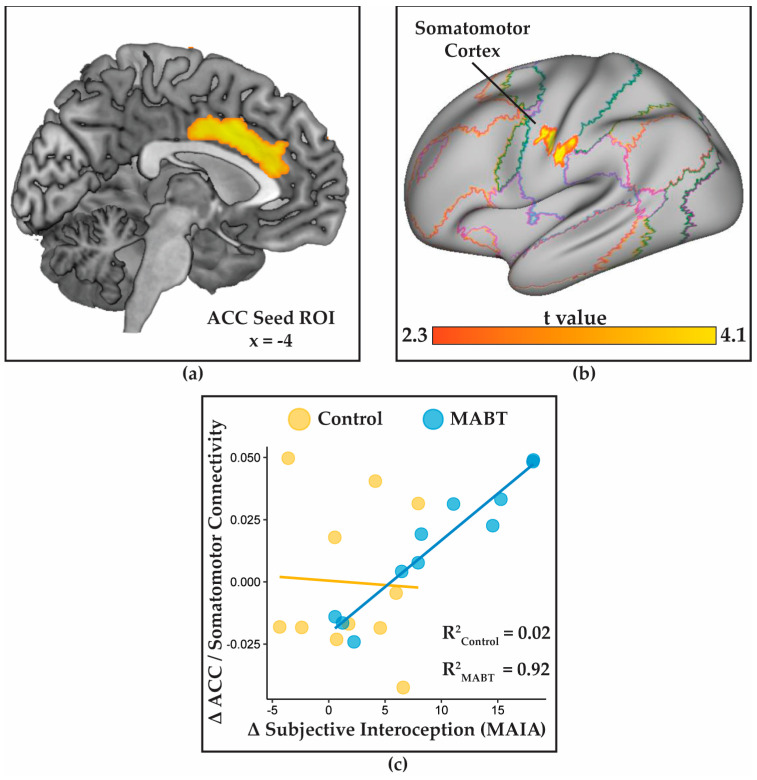
Psychophysiological interaction between changes in subjective interoception and changes in anterior cingulate cortex (ACC) connectivity: (**a**) the ACC seed ROI identified in [46]; (**b**) the somatomotor region demonstrating increased connectivity to the ACC ROI as a function of changing subjective interoception (MAIA scores); (**c**) the relationship between changes in subjective interoception (MAIA scores) and ACC–somatomotor connectivity. As the somatomotor ROI was identified using only within-group variance in the MAIA, the lines of best fit and R^2^ values are calculated separately for the MABT and control groups.

## Data Availability

All study analysis scripts and deidentified self-report data, as well as the insula ROI and extractions and connectivity change scores, are available at https://osf.io/ctqrh/ (accessed on 4 December 2018). Neuroimaging data are available upon request and fulfillment of appropriate data transfer agreements.

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
