# Peer review of "Within-Person Modulation of Neural Networks following Interoceptive Awareness Training through Mindful Awareness in Body-Oriented Therapy (MABT): A Pilot Study"

_brainsci, 2023, doi:10.3390/brainsci13101396_

Round 1
Reviewer 1 Report
The study focuses on the changes in functional connectivity related to interoceptive training. Specifically, the connection between the insula and PFC (prefrontal cortex) has been established in existing literature as important for processing interoceptive information. The results affirm this and further implicate the ACC (anterior cingulate cortex) and somatosensory cortex. This is in line with existing models that propose these areas play roles in body awareness and interoceptive processes.
Mindfulness-Based Interoceptive Training (MABT) appears to be a specific mindfulness training protocol, focused on enhancing interoceptive awareness. While mindfulness practices in general have been shown to impact brain connectivity, especially in the PFC and insula, MABT appears to be more specific to interoception. The identification of the somatosensory cortex in addition to the insula, is interesting and broadens the definition of interoception in accord with other models and findings.
In summary, while the findings offer exciting insights into the neural correlates of interoceptive training, they also prompt careful consideration of methodological limitations and the broader context of neuroscientific research on interoception and mindfulness. Future studies with larger, more diverse samples, combined with both objective and subjective measures, using this design approach could further solidify these findings.
-The High-Functioning sample used does raises questions about the applicability of the findings to clinical populations or those with specific disorders, but this has indeed been addressed in the discussion. On the other hand, limitations relating to the reliance on Self-Report Measures (like MAIA) should also be acknowledged.
-Referring to basic studies of interoception could be an advantage. This would help highlight the potential value of the findings, as well as enabling some further discussion on the different brain circuits that underlie the different aspects of interoception, and potential incostistencies in the literature.
https://doi.org/10.1038/s41586-023-05748-8
https://doi.org/10.1016/j.neuron.2019.12.027
https://doi.org/10.1038/s41423-023-01051-8
- - Supplementary figures S2 and S3 provide interesting additional insights that are discussed within the text, but should be referred to explicitly (at least once).
Author Response
Many thanks for your thoughtful comments. Based on your suggestion, we added some text to the limitations to emphasize need for more diverse sample and study in both clinical and non-clinical samples. We appreciate the suggestion of additional papers to provide a broader context for the varied proposed roles of the insula in homeostatic processes and have included them in the first first paragraph of the revised introduction. We agree that with the suggestion that the supplement appendices be referenced in the text and have amended the manuscript to do so, introducing an additional paragraph at the end of the Results section.
Reviewer 2 Report
Papers for review are so often flawed that it has been a delight to read one which is both so intellectually engaging and so conscientiously written!
I had the pleasure of reading an article that is both so intellectually engaging and so conscientiously written! I like this article because it manages to combine convincingly the theoretical part with data analysis.
This article is complete and informative, and written in a scientific tone. It also contains some original and unique aspects. The topic is relevant; the selected literature is appropriate and represents a complete review of the theoretical sources pertinent to the problem. Newly arisen questions are described. The literature is subjected to critical analysis. It is clear why this topic deserves a new discussion. The original theoretical perspectives and explanations are well grounded. The research problem under consideration is relevant. The article also contains original and unique aspects, such as an in-depth critical analysis of the literature or the introduction of new ideas to current literature. Recent scientific research that is relevant is included.
The structure, organization, coherence, and presentation of the material and the quality of the writing is excellent. The article contains a clear description of new research problems, and theoretical sources are properly referenced. All sections are relevant and informative, and the level of analysis is excellent. The implications and importance of the (sub)topics are critically discussed. The discussion added new insight to the current literature. The level of quality of the scholarship is excellent. Newly arisen issues are described. The text conforms to the appropriate academic style and is linguistically and grammatically correct. Overall, the article is very well written, well organized, and instructive to read.
Author Response
Many thanks for your thoughtful review and kind words. We appreciate your time and consideration in reviewing this paper.
Reviewer 3 Report
Thank you for the opportunity to review this manuscript. The aim of this study was to characterize changes in subjectively reported interoceptive awareness and associated interoceptive neural networks in response to an Mindful Awareness in Body-oriented Therapy (MABT) intervention compared to a passive, no treatment control condition among moderately stressed, community dwelling adults. The manuscript is clearly written and strongly justified according to precedents in the literature and relevance to understanding the neural basis to acquiring interoceptive awareness underlying wellbeing. The neuroimaging comparisons are clearly described in the intervention group and the findings are strong interest to researchers aiming to understand whether and how to improve behavioral and associated underlying neural markers of interoceptive function. The N is quite small but acknowledged by authors as a limitation, wisely placing these results in the realm of preliminary evidence in need of replication. Overall, I only see minor additions that may improve the manuscript for its publication.
Regarding the two groups, the intervention is clear as well as the contrasts used to show activation of interoceptive networks, i.e., Interoceptive and Exteroceptive attention (a breath tracking task vs. a visual tracking task). However, a little bit of detail as to what the “passive” no treatment control group did would be helpful to the reader. For example, did the passive condition likely involve activation of default networks that involve insular and ACC activity, and therefore underscore the importance of group differences or alternatively, can confound observation of between versus within group differences for example in the relationship between changes in subjective interoception and ACC connectivity?
The population is reported to have moderate to high anxiety and depression. It is highlighted that these conditions have been reported to show opposite influences on insular activity, specifically anxiety leading to increased and depression to decreased activations. Given the prevalence of both conditions, an additional note in the discussion as to the influence/relevance of these preexisting emotional conditions when discussing sparing the insula from widespread deactivation observed during the breath-monitoring task should be addressed.
Author Response
Many thanks for your review, the positive comments and your inquiries.
To clarify, the control group received no treatment, as indicated; likewise, as mentioned in the eligibility criteria (lines 155-157), they had to agree to not engage in any mind-body activities similar in orientation to the study intervention during the study timeframe (between baseline and post assessment). To make sure this is better understood, we removed the word "passive" in this revision and simply retained the no-treatment descriptor. We further clarify on revised line 172 that "The final sample included 11 participants in each experimental condition: MABT and a no-treatment Control group." We also clarify that the insula analyses (Aim 1) focused on the Interoception vs. Exteroception behavioral tracking task, but the connectivity analyses (Aims 2 and 3) focused on a sustained interoceptive attention task that did not contrast experimental conditions (revised lines 232-235 and 303-304).
The reviewer's second point about the like possibility of differential activity between groups affecting observations is well-founded, as the control group exhibited lower MAIA scores post training than the MABT: in a prior report (Farb et al., 2023, eNeuro), lower MAIA scores were associated with greater deactivation of the ACC during breath-focused attention. This 'flattening' of the activation profile which may have limited the ability to observe connectivity during interoceptive attention as reported here, and as illustrated by a lack of relationship between ACC/Somatosensory cortex and changes in MAIA score in Figure 3C. We speculate that this is not necessarily a confound, but rather part of the same profile-greater MAIA scores and preserved activity within the ACC may be necessary for effective integration of sensory signals from the somatosensory cortex. These ideas are now included in the discussion section on revised lines 506-511.
The reviewer's final point that the profile of spared insula deactivation is more in keeping with depression and than anxiety is well-taken, and is now mentioned on lines 568-571, with a caveat that 'pure' anxiety may present with a different profile than the mixed depression/anxiety symptoms reported here.